# Do factors across the World Health Organisation's International Classification of Functioning, Disability and Health framework relate to caregiver availability for community-dwelling older adults in Ghana?

Kofi Awuviry-Newton[1]*, Kylie Wales[1,2], Meredith Tavener[1], Julie Byles[1,3]

1 Priority Research Centre for Generational Health and Ageing, Faculty of Health and Medicine, University of Newcastle, Newcastle, Australia, 2 School of Health Sciences, The University of Newcastle, Newcastle, Australia, 3 Global Innovation Chair in Responsive Transitions in Health and Ageing, School of Medicine and Public Health, University of Newcastle, Newcastle, Australia

* newscous@gmail.com

**Data Availability Statement:** All relevant data are within the manuscript file.

## Abstract

### Introduction

In Ghana, the care needs of older adults in the later years has become a critical issue given population ageing and increased proportions of older adults with difficulties with functional abilities. However, factors related to caregiver availability is unknown. The purpose of this study was to examine how the World Health Organisation's International Classification of Functioning, Disability and Health (WHO-ICF) framework relates to caregiver availability for community-dwelling older adults in Ghana. This evidence will strengthen our understanding of the perceived unmet care needs of older adults in Ghana in Africa.

### Materials and methods

A hospital-based survey was conducted among 400 consecutively recruited older adult in-patients using a questionnaire at Komfo Anokye Teaching Hospital in southern Ghana. Multivariate logistic regression tested associations between caregiver availability and other factors as related to the WHO-ICF conceptual framework.

### Results

Eighty-six per cent of the participants reported having an available caregiver. In the final parsimonious model, the environmental factors were highly related to caregiver availability, seconded by personal factors, and then health conditions. Body function and structure, activity, and participation variables were not statistically significant. Overall, the variables that were associated with caregiver availability were age, being a widow, having a single chronic condition, being hardly understood by friends and family, receiving no neighbourhood support, and having 2–4 children. Interaction existed between being a widow and living as a couple in relation to caregiver availability.

**Funding:** The University of Newcastle International Postgraduate Research Scholarship (UNIPRS) and The University of Newcastle Research Scholarship Central 50:50 funded this study (UNRSC50:50). The Australian Research Council Centre of Excellence in Population Ageing Research (project number CE170100005) supported this research.

**Competing interests:** The authors have declared that no competing interests exist.

## Conclusions

Caregiver availability is associated with variables under the personal, health and environmental components of the WHO-ICF. Increased effort to strengthen the current and future welfare programs, including the health of older adults and their caregivers is relevant.

## Introduction

The population of people aged 60 years or older in Ghana is growing rapidly, mainly due to decreasing birth rates and delayed mortality [1]. The number of older adults in Ghana increased more than seven-fold from 213,477 (4.5%) in 1960 to 1,643,381 (6.7%) in 2010 [2]. The percentage of older adults is further expected to increase an additional 3.1% by 2050 [2]. Population ageing is associated with increases in service need, particularly for those who are living with functional difficulties [3]. Updated policies and programs are needed to respond to the current and future health needs of older adults in Ghana [4]. Among the essential service needs, requiring attention is long-term care. Long-term care refers to ranges of activities people engage in to care for others (including older adults) who are at risk of or experiencing functional difficulties. Long-term care aims to maintain some level of independence in line with their fundamental human rights, freedom and dignity. What is unknown is how ready people are to assume long-term caregiving roles for older adults with functional difficulties in the Ghanaian setting, where residential aged care facilities are non-existent.

As individuals age, they are likely to experience functional difficulties in carrying out daily activities. These activities may be fundamental for self-care, including feeding, bathing, toileting, grooming and dressing.[5]. Some activities are needed to live independently, including managing money and preparing meals. Moreover, functional activities may include social participation, including attending meetings [6].

In spite of increasing longevity worldwide, people may spend much of their later lives with some degree of functional difficulties [7]. In 2017, for instance, global life expectancy was 73 years, but healthy life expectancy was only 63 years; meaning 10 years of life was spent in poor health [8]. In sub-Saharan Africa, it is reported that the majority of older adults cannot perform everyday activities without assistance [9–11]. Specifically, a study in Ghana reported that about 90% of older adults require assistance in completing the things they value, including being able to visit friends [12]. Furthermore, the WHO estimated that about 50% of Ghanaian older adults between 65 and 75 years require assistance in functional activities [13]. The assistance older adults need to carry out functional activities is often provided by caregivers [14].

Globally, factors predicting caregiver availability for older adults with functional difficulties in formal aged care institutions have been investigated. The factors predicting caregiver availability for older adults in formal aged care institutions are usually older adults' advanced age, difficulty with self-care activities, without a partner, wealth, and those with dementia, advanced tumour or post-stroke [15, 16]. Other factors including being a woman, availability of family member, infrastructural investment, and insurance for long-term care [17, 18]. To the best of our knowledge, few studies on caregiver availability exist on community-dwelling older adults globally, with none existing on older adults in Ghana. Moreover, no study has employed a holistic framework of older adults' functioning to study caregiver availability. Globally, evidence suggests a variation of caregiver availability for community-dwelling older adults in terms of gender, marital status, and socio-economic status [19]. Specifically, older men without a partner lack caregivers in comparison with older women. Moreover, older

women were more likely to have an adult child as a caregiver than older men, whereas those from wealthy families do not have difficulty getting a caregiver [19]. In the United States of America, being a woman, not married, living alone, being older than 85 years, and reporting health as poor are more likely to perceive the availability of a caregiver [20].

Similarly, other evidence suggests a weakening desire of caregivers availing themselves for the care of community-dwelling older adults. For instance, a study reported a significant association existing between advanced age, functional ability and caregiver availability [21]. This study reported that older adults needing mobility assistance are less likely to have a primary caregiver. The older adults are 78% less likely to have a primary caregiver compared to the adults aged 40 to 59 years. There was no significant association found between older adults' gender and caregiver availability. In Nigeria, about 20% of community-dwelling older adults with functional difficulties do not receive care from either family members or friends (informal caregivers) [22]. One study on formal caregiving determined that the main factor that influenced the use of caregiving services was financial capacity. This study was conducted in China and did not include informal caregiving [15]. Given that there are limited studies exploring disability and caregiver availability, and none conducted in Ghana, it is timely to estimate the relationship between physical disability and caregiver availability for older adults.

In conducting such research, the use of a holistic exploratory and analytical framework to understand the factors associated with caregiver availability for community-dwelling older adults will prevent the probability of methodological individualism [23]. This is relevant in facilitating an understanding of the effective approach in offering long-term aged care for older adults at risk of functional decline [3].

In Africa, mostly Ghana, qualitative evidence describes factors that motivate people to care for community-dwelling older adults. These reasons are mainly caregiver-related factors, in mostly citing reciprocity **as** the reason for elder care [24–28]. For instance, caregivers reported that they provide care because their parent cared for them when they were infants [25, 29]. Other reasons that reportedly affect caregiving availability include caregivers' change in circumstances including the death of a spouse, availability of other family members, the proximity of children to older adults, not having a sister among siblings, and not having children [24–27, 30]. For instance, an adult child who stays close to the older adult family member becomes a caregiver [25]. Moreover, a family relative, who has no argument against caring eventually, becomes a caregiver [24]. In some cases, the deteriorating health of older adults as a result of disability or frailty influences the decision of people to become caregivers [26, 29, 31]. It appears that people become caregivers to escape the punishment they perceive may be meted on them by God, mainly if they do not provide care [25, 27, 28, 30]. Although this literature reports the influencing factors determining caregiver availability, they lack detail on how these factors together predict caregiver availability. Caregiver availability for community-dwelling older adults cannot be adequately understood without considering the circumstances in which older adults live, work and age, and other social supports systems as well as the interactions that may exist between them. To account for these personal and contextual factors, we employ the World Health Organisation's International Classification of Functioning, Disability and Health (WHO-ICF) framework to explore factors that may predict caregiver availability for older adults in Ghana.

The WHO-ICF offers a common language that can be used to describe and understand a person's health in terms of function and disability [32]. Function, as defined by the WHO-ICF, relates to all body functions, activities and participation [32] On the other hand, disability encompasses impairments, activity limitations and participation restrictions [32]. The WHO-ICF acknowledges the interaction between determinants of health and disability and, personal and environmental factors (including caregiver availability) [32, 33]. This framework

is also consistent with the World Health Organization [34] Framework for Healthy Ageing, which considers the dynamic interaction between the intrinsic capacity of the individual (all their physical and mental attributes) and the influence of physical and social environments that enable or limit their ability to achieve the goals they value. Accordingly, factors that contribute to or explain caregiver availability for older adults may include environmental, health status, community, and network factors, as well as their intrinsic capacities of older adults. Employing the WHO-ICF framework to explore caregiver availability may increase our understanding of the factors determining the current and future long-term care unmet needs of older adults. The current study models factors across the WHO-ICF framework and determines how they predict caregiver availability for older adults in Ghana.

Caregiver availability is an environmental factor that may be affected by intrinsic personal factors such as gender, health condition, body function and structure, and activity and participation. A study that explored the relationship between caregiver availability and health reported that if older adults did not perceive themselves to be isolated and that they had support from a caregiver, their health and overall functioning is likely to be better [35]. It can also be inferred that when there is perceived absence of caregiver availability, older adult's involvement in life activities or participation in things they value may be curtailed because of lack of instrumental support (absence of caregivers) potentially worsening their health [36]. To the best of our knowledge, there is a paucity of information on the relationship between physical disability and caregiver availability among older adults, and particularly in Africa where formal care systems are not widely available. This current study will address this gap in the literature by exploring caregiver availability in the context of the components of the WHO-ICF. This information will enhance our understanding of the holistic factors determining older adult's functioning level. We hypothesized that the higher the needs across the components of the WHO-ICF, the more older adults would need a caregiver.

## Materials and methods

### Study design

A cross-sectional survey design was employed.

### Study area

The survey was conducted in Komfo Anokye Teaching Hospital (KATH) located in Kumasi, the Regional Capital of Ashanti Region with a population of 4,780,249 as of 2010 [37]. KATH is accessible to a wide range of people from 13 regions of the 16 administrative regions of Ghana.

### Study sampling

The required sample size for the study was 400 participants. To achieve this sample, all older adults admitted to the hospital between August, and December 2018 were eligible to participate if they met the following inclusion criteria; 1) older adults (60 years or over), 2) admitted to the hospital for any health reason or frailty, 3) stayed a minimum of one night, 4) not be seriously ill or unconscious and, 5) provided consent. This study sampled participants from a hospital setting, unlike the World Health Organisation study on AGEing and adult health (WHO-SAGE) that sampled older adults from their homes who were less likely to have needs for care [38].

Participant recruitment was completed on randomly selected days (4 days per week) over eight hours to increase the likelihood of participants who were admitted being offered the

opportunity to participate. The study employed consecutive sampling techniques to recruit participants who met the eligibility criteria.

## Data collection

Nurses determined the eligibility of participants for the study based on the inclusion criteria. A trained research assistant approached potential participants, provided them with the participant information statement and explained the study to them. The research assistant returned after 24–48 hours to follow up and obtained the informed consent of potential participants. The primary author then approached the consenting participants to decide the time and place to participate in the study during their hospitalisation. Participants completed a survey, where the researcher read the questions to solicit responses from participants who could not read. Participants answered questions related to their socio-demographic information, their current hospital admission, and current functional ability using WHO-SAGE questionnaire, as well as support and care (see S1 Appendix for survey question). All communication was in "Twi", the dominant language of Ghana.

## Dependent variable

**Caregiver availability.**    In the survey, participants were asked "Do you have a caregiver?" on a dichotomous response ("yes-1" and "no-2"). For logistic regression, "no" was coded as "0" and "yes" was coded as "1" and considered a dependent variable.

## Independent variables

**Personal Factors (PF).**    Personal factors are contextual factors that are intrinsic to the individual, which may influence how a person functions [33]. We hypothesized that personal factors would predict caregiver availability, in the context of other needs. For instance, older adults living with family are expected to have caregivers available within their home. In contrast, adults living alone may not have a caregiver. According to the WHO-ICF, PF included age, sex, marital status, education, religion, residence, living arrangement, and employment status. In this study, age was measured as a continuous variable. Sex was measured on a categorical nominal variable "male" or "female". Marital status was measured with "never married", "married/cohabiting", "separated/divorced", and "widowed". Marital status was categorised as "single", "separated or divorced", "married or cohabiting", and "widowed" for the purpose of chi-square analysis. In terms of education, the response categories were "no education", "at most junior high completed", "senior high completed" and "at least college/ pre-university completed". This was further categorised as "no education", "at most junior high completed", and "at least senior high school completed". Religion was initially measured as "none", "Christianity (including Roman-Catholics)", "Islam", and "traditional religion". However, for the purpose of the analysis, the "none" and "traditional religion" were combined as "none", with "Christianity" and "Islam" treated as separate categories. In terms of living arrangements, the categories were "alone", "as a couple", "as a couple and with children". Employment status was categorised as "currently working" and "currently not working".

**Body function and structure (BFS).**    Body function and structure refers to the impairment or function level of body anatomy or physiology, which may affect overall health [33]. It was expected that older adults with impairment in BFS would have a greater need for care, and therefore be more likely to have a caregiver than those with nobody impairment. The survey **items** assessing the BFS w**ere** "what illness did the doctor/nurse diagnose you with . . .?" with responses related to body impairment including "injury" and "visual impairment".

Impairment was coded as "1" meaning "yes" if not present, and "0" meaning "no" if not present.

**Health Conditions (HC).**   Health condition refers to any illness or chronic condition that may affect older adults overall functioning [33]. We hypothesized that older adults living with illness or chronic condition will be more likely to need care, and therefore more likely to have a caregiver available to them. As a result, the question "what illness did the doctor/nurse diagnose you with. . .?" was used to explore older adults' HC. These conditions include chronic condition, communicable diseases and alcoholism. The health conditions mainly reported by participants were diabetes, stroke, ulcers, cancer, hypertension, kidney disease, asthma, heart disease, and lung diseases. Other health conditions reported were hernia, malaria, chronic alcoholism, heart failure, jaundice, atrial fibrillation, ganglia, urosepsis, haemorrhage, gastroenteritis, monoparesis, uremic encephalopathy, anaemia, urinary tract infection, cardiomyopathy, goitre, chronic left external capsular infarct, appendicitis, liver disease, pneumonia, intestinal lung disease, cellulitis, gangrene, hepatitis, cataract, fall injury, kidney disease, blindness, fibroid, angina, prostate enlargement, and neurological problem. Each of these variables were categorized as "1" meaning "yes" if the condition was present and "0" meaning "no" if the condition was not present. For the purpose of the analysis, these conditions were transformed into multi-morbidity variable coded as "no condition", "at least one condition" and "two or more conditions".

**Activity Limitation (AL): Disability.**   Activity limitation or disability refers to the difficulty older adults may experience in engaging activities necessary for life [33]. We hypothesised that older adults experiencing difficulty with everyday functional activities (disability), will have a caregiver available to them. Disability was assessed in terms of activities of difficulties on 24 items of daily living (ADL), instrumental activities of daily living (IADL), participation, and mobility. Difficulties were graded using ordinal response categories (none (0), mild (1), moderate (2), severe (3), extreme (4)) in response to the question "In the last 30 days, how much difficulty did you have with . . . . . . . . . . . . . . . . . due to health problems, injuries, mental or emotional problems**?"** Internal consistency of response across the 24 items in this study was assessed, and reliability was found to be high (Cronbach alpha = 0.94).

**Environmental Factors (EF).**   Environmental factors refer to the physical, social and attitudinal setting in which older Ghanaians undertake life activities [33]. We hypothesised that older adults with a more favourable and supportive social environment and that those with greater support networks would be more likely to have caregiver availability.

To assess the social environment, participants were asked "How many children do you have? The response was categorised as "none", "1–5", and "6 or more". They were also asked"-Where do you receive support?" with supports including neighbours, community, government, religious/members and non-governmental organisations, and responses categorised as "yes" if the support was present and "no" if no support from that source. However, in the analysis religious support was removed from the model as it is highly correlated with neighbourhood support and did not better predict caregiver availability. An emotional support variable, was "How many times during the past week did you spend time with someone who does not live with you, that is, you went to see them, or they came to visit you, or you went out together?". This question was measured **as** an ordinal categorical variable beginning from 0 to 7 or more but was categorised as "none", "1–5 times" and "6 or more times" for analysis. Perceived social support was measured using nine questions. An example of these questions was, "Does it seem that your family and friends, people who are important to you understand you? The response categories were "hardly ever, "some of the time", and "most of the time". For the purpose of analysis, a correlation matrix was conducted for the nine variables to access the level of relationship, and highly correlated variables were removed. The question, about

"family and friends understanding you" was included in the model building because it was a good predictor.

**Participation (P).** Participation restriction refers to challenges an older adult may experience in engaging in meaningful activities [33]. We hypothesised that older adults with difficulties participating will have caregivers available to care for them. This is based on a study reported that older adults find it difficult to participate in life if they do not have instrumental support [36]. The question asked to determine participation restriction level of participants was "about how often did you go to meetings of clubs, religious meetings or other groups that you belong to in the past week?" The question was measured on the response beginning from "none" to "seven or more". These responses were categorized as "none", and "at least once".

## Data analysis

Descriptive statistics were used to describe demographic information of study participants. To compare relationships between categorical variables, a Chi-square test was used. Bivariate and multivariable logistic regression was performed to assess any significant relationship between variables under the WHO-ICF components (independent variables) and caregiver availability (dependent variable). Logistic regression was used to estimate crude and adjusted odds ratios and 95% confidence intervals to test for associations between the dependent and independent variables. The regression used the *logit* function for the *dependent variable* and the categorical *independent variables*. Variables with potentially significant associations with caregiver availability (p-value of less than 0.2 on bivariate analyses) were tested in nested multivariable logistic regression models to control potential confounding variables. The P-value was set at less than 0.05 to determine statistical significance in the final models. Using the WHO-ICF framework, individual variables under each WHO-ICF component were modelled, and then other components were added to PF through EF to understand the association between different WHO-ICF components and caregiver availability. Post hoc analysis was conducted to determine the existence of interaction. STATA version 15 was used to manage the analysis.

## Ethical consideration

Ethical approval (H-2018-0163) for this analysis was obtained from The University of Newcastle in Australia in keeping with the Declaration of Helsinki. Informed consent was obtained from the director of Komfo Anokye Teaching Hospital and study participants. Anonymity and confidentiality were ensured.

## Result

Table 1 illustrates the demographic characteristics of the participants. Most participants were women (51%), and the mean age was 72 years. Moreover, about 82% of participants had a caregiver available. (Refer to Table 1 for further information).

### Bivariate analysis of caregiver availability across the WHO-ICF components

Factors that had p<0.2 on bivariate analysis were considered for further analysis (see S1 Table). Sex, education, residence, employment, injury, disability score, speaking with someone on the telephone, government support, and meeting attendance were not significantly associated with caregiver availability. Participants with no caregiver available had no form of non-government support (100%), whereas only 2.9% of those with a caregiver did. Non-governmental support could not be included in the analysis because of a zero cell count.

**Table 1. Demographic characteristics of older adults, and caregiver availability.**

| Demographic Characteristics (N = 400) | N (%) |
|---|---|
| Age *(mean, SD)* | 71.3±8.42 |
| **Sex** | |
| Male | 196 (49.0 |
| Female | 204 (51.0) |
| **Marital status** | |
| Single/separated/divorced | 58 (14.5) |
| Currently married/cohabiting | 212 (53.0) |
| Widowed | 130 (32.5) |
| **Education** | |
| No education | 128 (32.0) |
| At maximum junior high completed | 209 (52.3) |
| At least senior high completed | 63 (15.8) |
| **Religion** | |
| None | 27 (6.75) |
| Christianity | 331 (82.8) |
| Islam | 42 (10.0) |
| **Residence** | |
| Rural | 227 (56.8) |
| Urban | 173 (43.3) |
| **Living arrangement** | |
| Alone | 50 (12.5) |
| With couple | 100 (25.0) |
| With couple and children | 250 (62.5) |
| **Employment status** | |
| Currently working | 156 (39.0) |
| Currently not working | 244 (61.0) |
| **Availability of a Caregiver?** | |
| Yes | 344 (86.0) |
| No | 56 (14.0) |

## Modelling caregiver availability according to the WHO-ICF framework

Detailed Multivariable regression results, for Models (1–6), based on the WHO-ICF framework, and are present in S2 Table. In Model 1, after adjusting for all significant PF for every one-year additional increase in age, the odds of caregiver availability increased by 4% (p<0.2). Participants who were single separated and divorced, and the widowed were 50% (p<0.2) and 55% (p<0.05) respectively less likely to have a caregiver available than the married and cohabiting. There was no statistically significant association between religion and caregiver availability (p>0.2). Participants living alone were 58% less likely to have a caregiver available compared to those living as a couple and with children.

In Model 2, all PF that remained significant in Model 1 were included along with BFS variables where PF<0.2 on the bivariate analysis. However, none of the BFS variables was significantly associated with caregiver availability in this model. All significant personal factor variables from Model 1 were retained in Model 2.

In Model 3, all variables that remained significant in Model 2 were adjusted with HC variables where p<0.2 on the bivariate analysis. In this model, after adjusting for PF those who have any two or more conditions were 3.98% times more likely to have a caregiver available than the participants living with any one condition.

In Model 4, all variables that remained significant in Model 3 were included with the disability score (AL variable). However, disability scores were not statistically significantly associated with caregiver availability. Likewise, in Model 5, no P variable was significant, and thus all factors from Model 3 were retained in Model 4 through to Model 5.

In Model 6, the final model, all variables that remained significant in Model 5 were included with variables from the EF component that were significant at the bivariate analysis stage (see Table 2). In this final model, after adjustment for all other WHO-ICF components, there was little change in the statistically significant associations comparing crude and fully adjusted models. For every one year additional increase in age, the caregiver availability increased by 3%, but this was not significant at the p<0.05. For marital status, the widowed were 51% (p<0.05), less likely to have a caregiver available compared to the married or cohabiting. There was no statistically significant association between being single, separated or divorced and caregiver availability (p>0.05), compared to the married or cohabiting. Participants with two or more health conditions were 258% (p<0.05) more likely to have a caregiver compared to those having one health condition. There was a slight attenuation in the size of the effect of this component between crude (OR 4.10) and the final model (OR 3.58). In terms of perceived support, those who reported that they hardly feel that family and friends understand them were 81% (p<0.001) less likely to have a caregiver compared to those who reported that they were understood some of the time. Participants who had not spent time with someone who does not stay with them were 62% (p<0.05) less likely to have a caregiver compared to those who spent time with someone for 1–5 times for the past week. More so, participants who reported receiving no neighbourhood support were 61% (p<0.01) less likely to have a caregiver available compared to those who receive neighbourhood support. Finally, those who reported having 2 to 4 children were 52% (p<0.05) less likely to have a caregiver less than in those who reported having at least five children.

A post hoc analysis to test for an interaction between marital status, living arrangement and caregiver availability identified that interaction existed between being widowed, marital status, and living as a couple (p = 0.048). This implies that the relationship between widowed and living with others (as a couple) appears to be different from the relationship between single, separated or divorced and living alone (p = 0.339); between being single, separated, or divorced, and living as a couple (p = 0.222); and between widowed and being single, divorced, or separated and living alone (p = 0.568). This implies that if not for the interaction from living as a couple, being a widowed would have been 0.51 less likely to have a caregiver than odds of caregiver availability in the married or cohabiting.

## Discussion

Through the application of the WHO-ICF, our study demonstrates that caregiver availability for older adults is influenced by environmental, personal and, health-related factors. Factors that do not influence caregiver availability are body function and structure, activity, and participation variables. The results suggest that not all components of the WHO-ICF influence caregiver availability for older adults living in Ghana.

Modelling, according to the WHO-ICF, we noticed a few small variations in the odds ratios. In the final model, health conditions factors were adjusted for other factors in the WHO-ICF; some attenuation of the odds ratio was observed. This finding implies that along with chronic conditions, variables under these other components of the WHO-ICF together influence caregiver availability. These findings emphasize that interventions for older adults with chronic conditions should also aim to strengthen older adults' intrinsic wellbeing, improve health and make conducive environments that facilitate participation to assist older adults in meeting

**Table 2. Modelling caregiver availability of community-dwelling older adults according to the WHO-ICF framework.**

| Variables across WHO-ICF components | COR, C.I | AOR (Model 6-PF+BFS+HC+AL +PR+EF) |
|---|---|---|
| **PERSONAL FACTORS** | | |
| **Age *(mean, SD)*** | 1.04 (1.00, 1.07)* | 1.03 (0.99, 1.08)* |
| **Gender** | | |
| Male (vs female) | 01.04 (0.59, 1.83) | |
| **Marital status** | | |
| Single/separated/divorced (vs married/cohabiting) | 0.40 (0.18, 0.88)** | 0.67 (0.27, 1.68) |
| Widowed (vs married/cohabiting) | 0.46 (0.24, 0.87)** | 0.49 (0.24, 1.00)** |
| **Education** | | |
| No education (vs at most junior high completed) | 0.75 (0.41, 1.37) | |
| At least senior high completed (vs at most junior high completed) | 1.47 (0.58, 3.73) | |
| **Religion** | | |
| None (vs Christianity) | 0.45 (0.18, 1.12)* | |
| Islam (vs Christianity) | 1.50 (0.51, 4.39) | |
| **Residence** | | |
| Urban (vs Rural) | 1.11 (0.63, 1.97) | |
| **Living arrangement** | | |
| Alone (vs With couple and children) | 0.32 (0.16, 0.65)*** | 0.83 (0.34, 1.99) |
| With couple (vs With couple and children) | 1.10 (0.53, 2.30) | 1.08 (0.48, 2.43) |
| **Employment status** | | |
| Currently working (vs Currently not working) | 0.76 (0.43, 1.35) | |
| **BODY FUNCTION AND STRUCTURE** | | |
| **Visual impairment** | | |
| Yes (vs No) | 0.65 (0.07, 5.90) | |
| **Injury** | | |
| Yes (vs No) | 1.38 (0.52, 3.66) | |
| **HEALTH CONDITION** | | |
| **Multi-morbidity** | | |
| no condition (vs Any 1 condition) | 1.53 (0.82, 2.86) | 1.67 (0.83, 3.68)* |
| Any 2 or more conditions (vs Any 1 condition) | 4.10 (1.54, 10.9)*** | 3.58 (1.24, 10.3)** |
| **ACTIVITY LIMITATION** | | |
| Disability score (mean, SD) | 1.00 (0.98, 1.00) | |
| **PARTICIPATION** | | |
| **Often times you attend meetings (past week)** | | |
| At least once (vs None) | 1.03 (0.22, 4.86) | |
| **ENVIRONMENTAL FACTORS** | | |
| *Perceived Support* | | |
| **Family and friend understand you** | | |
| Hardly ever (vs Some of the time) | 0.20 (0.10, 0.41)*** | 0.19 (0.08, 0.43)**** |
| Most of the time (vs Some of the time) | 0.70 (0.33, 1.49) | 0.69 (0.30, 1.58) |

(*Continued*)

**Table 2.** (Continued)

| Variables across WHO-ICF components | COR, C.I | AOR (Model 6-PF+BFS+HC+AL+PR+EF) |
|---|---|---|
| *Emotional support* | | |
| **Spent time with someone who does not live with you (past week)** | | |
| None (vs 1–5 times) | 0.33 (0.16, 0.68)*** | 0.38 (0.17, 0.87)** |
| 6 or more times (vs 1–5 times) | 1.17 (0.60, 2.29) | 0.69 (0.32, 1.49) |
| **Often time you spoke with someone via telephone (past week)** | | |
| 1–5 times (vs none) | 0.97(0.52, 1.82) | |
| 6 or more times (vs none) | 0.77 (0.35, 1.68) | |
| **Neighbours/community support** | | |
| No (vs Yes) | 0.42 (0.21, 0.85)* | 0.39 (0.19, 0.80)*** |
| **Government support** | | |
| No (vs Yes) | 1.01 (0.51, 2.00) | |
| **Number of children** | | |
| At most one child (vs 5 or more) | 0.34 (0.14, 0.82)* | 0.90 (0.31, 2.58) |
| 2–4 (vs 5 or more) | 0.39 (0.21, 0.73)** | 0.48 (0.24, 1.00)**' |

PF- personal factors; BFS- body function and structure; H- health condition; E- environmental factors; AL-activity limitation; P-participation; AJHS- At least junior high completed; WCC- with couple and children

*p-value < 0.2

**p-value < 0.05

***p-value < 0.01

****p-value<0.001; COR-crudes odds ratio, AOR-adjusted odds ratio.

their unmet care needs. This study highlights that many variables within the environmental factor of the WHO-ICF framework predict caregiver availability for older adults. Specifically, those participants who perceived that their friends and family hardly ever understand them had the lowest levels of caregiver availability. Moreover, those who reported they hardly spent time with someone who does not live with them had low levels of caregiver availability. Given the weakening levels of extended family in Ghana [39–41] and complaints of elder neglect [42], when there is a misunderstanding between participants and significant others (friends or own family (couple and children)), having a caregiver available may be problematic. Disagreement with participants often leads to dissociation from social relationships, which in turn may lead to unavailability of caregivers. This discussion indicates how availability and adequate environmental factors are to increase older adults' caregiver availability. This draws further attention to the need to introduce programs that focus on bridging the communication gap and helps connect older adults to their caregivers and express their care needs.

Similarly, participants who received no neighbourhood or community support had low levels of caregiver availability. This finding indicates that once neighbours do not support participants, participants' unmet support and care needs will increase because of the declining extended family system in Ghana. This might also imply that participants' partners or children may need to support themselves or do not live with older adults in the same home. This circumstance has been reportedly influenced by migration and modernisation [39].

In addition, having two to four children was associated with having lower levels of caregiver availability compare to having at least five children. This finding indicates that the probability of one child among five or more children to live with participants to provide care is likely. This study finding implies that in the current economic instability, and other pursuits in Ghana where parents may not want to give birth to more than four children [2] places emphasis on the need for strengthening public social welfare systems to compensate for the caregiving loss as a result of social change variables.

In this study, many of the older adults had caregivers available to them; however, caregiver availability did not significantly increase with age. This finding is consistent with studies that found a non-linear relationship with caregiver availability [20, 21, 43], reporting that participants older than 85 years had lower levels of perceived caregiver available to them. More research is needed into the relationship between age and caregiver availability.

Moreover, widows had lower levels of caregiver availability than married or cohabiting. This finding is consistent with the current literature [20, 44].

In this current study, the older adults' reason for admission to hospital was mainly due to accident or falls, rather than illness. Participants' partners or family members may have been present due to participants' injury, which may have needed immediate short-term support and care. Participants with two or more health condition had higher levels of caregiver availability. This finding is similar to caregiver availability for older adults in a formal residential care [15]. This finding implies that older adults having multiple health conditions may have caregivers available, which may be due to their increased need and not necessarily be due to reciprocity.

While not directly evident from our study, the finding that widowed older adults have lower levels of caregiver availability implies that there are many single older adults who do not have a caregiver and may be vulnerable to neglect, exploitation, or abuse. This draws attention to the need to strengthen the social welfare systems to cover this vulnerable population. Identifying a possible caregiver and deliberating on likely future caregiving needs with family members and friends is a significant health promotion and maintenance activity for older adults. Moreover, the lack of strength in the significant relationship between age and caregiver availability implies that increasing age in Ghana does not necessarily mean they will have a caregiver available to them. This study also suggests that social and healthcare service should be supportive of caregivers of older adults, and supplementary to the care they are able to provide. More research is needed to understand the functional status of Ghanaian older adults to understand their health and social care needs. Moreover, research employing the WHO-ICF to explore caregivers' availability for older adults from the perspective of caregivers, will elucidate our understanding of the unmet caregiving needs of older adults in Ghana in Africa.

The strength of this study lies in its application of the WHO-ICF to examine factors associated with caregiver availability and the use of multivariate logistic regression, where the variables were adjusted for confounding and interaction. This study has, therefore contributed to the validation of this framework in the context of caregiver availability from the perspective of older adults. There are certain limitations that need to be acknowledged. There are, however, some limitations. These include the relatively small sample size, which may have limited the power to identify associations and interactions. We also did not measure the quality or intensity of caregiving. A further limitation is that some components of the WHO-ICF did not have enough variables to assess the domain in full, which may have affected the ability to determine the relationship between the WHO-ICF components and caregiver availability. Future research should include more variables or factors under each component should be conducted to confirm the association revealed in this study.

## Conclusion

A significant number of caregivers are available for older adults in Ghana. However, a substantial number of widowed older adults, who needed to support the most, do not have caregivers available to them. Caregiver availability is associated with variables under three components of the WHO-ICF, emphasising the personal, health, and environmental factors associated with caregiver availability for older adults. Increased effort to strengthen the current and future social support, health and personal needs of older adults are needed to improve their wellbeing. Based on the findings of this study, government intervention should target widowed older adults living alone, and those without strong relationships with family or friend for they are less likely to have caregivers. Moreover, since having 2–4 children were less likely to receive care compared to having five children, there should be financial, health and emotional support for those who would need to avail themselves for the care for other adults. The current study has highlighted the importance of neighbours in the caregiver availability. This finding draws attention to the need to educate the community on the care needs of older adults to increase the possibility of a person assuming the care, especially when the immediate family are not available.

Future research should undertake an in-depth qualitative exploration of the WHO-ICF so that older adults' experiences with caregiver availability are understood.

## Supporting information

**S1 Appendix. A survey questionnaire.**
(DOCX)

**S1 Table. Bivariate analysis of caregiver availability across the WHO-ICF components.**
(DOCX)

**S2 Table. Full modelling of caregiver availability according to the WHO-ICF framework.**
(DOCX)

## Acknowledgments

We are grateful to Prof Mel Gray for her initial contact with the primary author and the Research Center for Generational Health and Ageing for making educational resources available for data analysis and write-up.

## Author Contributions

**Conceptualization:** Kofi Awuviry-Newton, Kylie Wales, Meredith Tavener, Julie Byles.

**Formal analysis:** Kofi Awuviry-Newton.

**Investigation:** Kofi Awuviry-Newton.

**Methodology:** Kofi Awuviry-Newton, Kylie Wales, Meredith Tavener, Julie Byles.

**Project administration:** Kofi Awuviry-Newton.

**Software:** Kofi Awuviry-Newton.

**Supervision:** Kylie Wales, Meredith Tavener, Julie Byles.

**Validation:** Kylie Wales, Meredith Tavener, Julie Byles.

**Writing – original draft:** Kofi Awuviry-Newton.

**Writing – review & editing:** Kylie Wales, Meredith Tavener, Julie Byles.

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
