## [Decision Letter · Decision Letter 0]

22 Jan 2020

PONE-D-19-33768

Do factors across The World Health Organisation International Classification of Functioning, Disability and Health framework explain perceived caregiver availability for community dwelling older adults in Ghana?

PLOS ONE

Dear Dr Awuviry-Newton,

Thank you for submitting your manuscript to PLOS ONE. After careful consideration, we feel that it has merit but does not fully meet PLOS ONE’s publication criteria as it currently stands. Therefore, we invite you to submit a revised version of the manuscript that addresses the points raised during the review process.

We would appreciate receiving your revised manuscript by Mar 07 2020 11:59PM. To enhance the reproducibility of your results, we recommend that if applicable you deposit your laboratory protocols in protocols.io, where a protocol can be assigned its own identifier (DOI) such that it can be cited independently in the future. For instructions see: http://journals.plos.org/plosone/s/submission-guidelines#loc-laboratory-protocols

We look forward to receiving your revised manuscript.

Kind regards,

Frédéric Denis, Ph.D.

Academic Editor

PLOS ONE

Journal Requirements:

3. During our internal checks, the in-house editorial staff noted that you conducted research or obtained samples in another country. Please check the relevant national regulations and laws applying to foreign researchers and state whether you obtained the required permits and approvals. Please address this in your ethics statement in both the manuscript and submission information.

4. Please include additional information regarding the survey or questionnaire used in the study and ensure that you have provided sufficient details that others could replicate the analyses. For instance, if you developed a questionnaire as part of this study and it is not under a copyright more restrictive than CC-BY, please include a copy, in both the original language and English, as Supporting Information.

5. Please amend either the title on the online submission form (via Edit Submission) or the title in the manuscript so that they are identical.

The University of Newcastle International Postgraduate Research Scholarship (UNIPRS) and

The University of Newcastle Research Scholarship Central 50:50 funded this study

(UNRSC50:50).

The Australian Research Council Centre of Excellence in Population Ageing Research (project

number CE170100005) supported this research.

No funding

7.  In your Data Availability statement, you have not specified where the minimal data set underlying the results described in your manuscript can be found. PLOS defines a study's minimal data set as the underlying data used to reach the conclusions drawn in the manuscript and any additional data required to replicate the reported study findings in their entirety. All PLOS journals require that the minimal data set be made fully available. For more information about our data policy, please see http://journals.plos.org/plosone/s/data-availability.

Reviewers' comments:

Reviewer's Responses to Questions

**Comments to the Author**

1. Is the manuscript technically sound, and do the data support the conclusions?

Reviewer #1: Partly

Reviewer #2: Yes

2. Has the statistical analysis been performed appropriately and rigorously? 

Reviewer #1: No

Reviewer #2: I Don't Know

3. Have the authors made all data underlying the findings in their manuscript fully available?

Reviewer #1: Yes

Reviewer #2: Yes

4. Is the manuscript presented in an intelligible fashion and written in standard English?

Reviewer #1: Yes

Reviewer #2: Yes

5. Review Comments to the Author

Reviewer #1: WHO-ICF is an essential topic among researchers. However, I think that the vital elements are not provided to understand the findings. Also, the introduction and discussion would not be strong arguments. It requires additional literature reviews.

ABSTRACT

The background is missing.

INTRODUCTION:

1. The introduction needs to have extensive literature reviews linking to what the authors examined. Especially any relationship among the WHO-ICF and caregiver availability is not clear. Please describe how the authors use the WHO-ICF model in caregiver availability well. Why did you use the WHO-ICF? Instead, you can use the social support theory.

2. Please explore more about the relationship between physical disability and caregiver more in details

3. What are your hypotheses?

Methods.

3. What is the terminally ill? Please define it.

4. understudy sample, Did you collect the data from the participants of the WHO on AGEing and adult health study? It is not clear.

5. Please include the references for the measurements. Some of them came from the articles. How about the rest?

6. Data analysis. Please write more details about the software, code to obtain the odds ratios.

7. The order of subsection n the method section is not correct.

8. Please explain the relationships between the selected variables and subdomains in the WHOICF.

Discussion

The first paragraph is repetitive, like results. Please summarize it.

Please exam your findings based on your hypotheses.

Since Ns of variables per domain is small. Instead of using the WHO-ICF, it would be better to examine the variable itself. Like marital status, chronic conditions. Your findings are similar to previous studies. Please explain why and how it happened. Please expand your conclusions to your unique area for the implication section.

Reviewer #2: This is an interesting manuscript that is worthy of publication.

A few minor points that need to be addressed:

The first sentence of the second paragraph of the intrduction does not make sense "...they need to do to live and those they most cherish." In addition, the reference that goes with this and the examples of self care is inappropriate (Shiel) and does not illustrate the point. 3rd paragraph of the introduction - the Heaton reference is not listed. Last paragraph of the introduction - the WHO 2002 references should read 2005.

In the methods section, it is unclear as to whether the participants had to come back into the hospital to complete the survey, or if it was completed during their hospitalisation. If they had to come back, this is a bias because more participants with caregivers might be expected to have been able to arrange to come back than those without, i.e. the sample will be biased towards those with support.

In the results, in the third sentence of the paragraph 'Bivariate analysis' the sentence "Participants with no caregiver available had some no form of non-government support" does not make sense. In the paragraph commencing 'In Model 6...', the sentence "Participants living alone was not statistically significant" needs completing.

Discussion last sentence of third paragraph repetition of the word 'modernisation'.

References: the Ghana statistical Service reference is not cited in the text.

6. PLOS authors have the option to publish the peer review history of their article (what does this mean?). If published, this will include your full peer review and any attached files.

Reviewer #1: No

Reviewer #2: No

---

## [Author Response · Author response to Decision Letter 0]

21 Feb 2020

Dear Dr Frederic Dennis, 

We are very grateful for the privilege to revise the manuscript. In this letter, we have responded to the reviewers’ and your comments. 

Journal Requirements:

Response: 

We have ensured that the manuscript meets PLOS ONE style requirement. 

Response: 

A caption for the Supporting Information files have been provided. 

3. During our internal checks, the in-house editorial staff noted that you conducted research or obtained samples in another country. Please check the relevant national regulations and laws applying to foreign researchers and state whether you obtained the required permits and approvals. Please address this in your ethics statement in both the manuscript and submission information.

Response: 

We obtained two ethics approval, one from Ghana and one from The University of Newcastle. The study has been conducted within the various national regulations.

4. Please include additional information regarding the survey or questionnaire used in the study and ensure that you have provided sufficient details that others could replicate the analyses. For instance, if you developed a questionnaire as part of this study and it is not under a copyright more restrictive than CC-BY, please include a copy, in both the original language and English, as Supporting Information.

Response: 

Survey instruments has been attached. 

5. Please amend either the title on the online submission form (via Edit Submission) or the title in the manuscript so that they are identical.

Response: 

We have amended the title for identical purpose

The University of Newcastle International Postgraduate Research Scholarship (UNIPRS) and

The University of Newcastle Research Scholarship Central 50:50 funded this study

(UNRSC50:50).

The Australian Research Council Centre of Excellence in Population Ageing Research (project

number CE170100005) supported this research.

No funding

Response: 

We have removed the funding statement in the manuscript. We will like you to update our funding statement as Funding Statement “The University of Newcastle International Postgraduate Research Scholarship (UNIPRS) and The University of Newcastle Research Scholarship Central 50:50 funded this study (UNRSC50:50). 

The Australian Research Council Centre of Excellence in Population Ageing Research (project number CE170100005) supported this research.”

7. In your Data Availability statement, you have not specified where the minimal data set underlying the results described in your manuscript can be found. PLOS defines a study's minimal data set as the underlying data used to reach the conclusions drawn in the manuscript and any additional data required to replicate the reported study findings in their entirety. All PLOS journals require that the minimal data set be made fully available. For more information about our data policy, please see http://journals.plos.org/plosone/s/data-availability.

Response: 

We have made amendment to the Data Availability statement. As a result upload of the dataset is not required for this submission. 

Response to reviewers 

Response to reviewers: Do factors across the World Health Organisation International Classification of Functioning, Disability and Health framework explain perceived caregiver availability for community-dwelling older adults in Ghana?

Reviewer #1 

WHO-ICF is an essential topic among researchers. However, I think that the vital elements are not provided to understand the findings. Also, the introduction and discussion would not be strong arguments. It requires additional literature reviews.

Authors appreciate reviewer’s acknowledgment of the essential need of WHO-ICF framework, and other advice. Our responses below show how we have addressed these issues.

ABSTRACT, The background is missing.

We have provided additional information in the abstract introduction. Please see bold text in introduction. 

INTRODUCTION:

1. The introduction needs to have extensive literature reviews linking to what the authors examined. Especially any relationship among the WHO-ICF and caregiver availability is not clear. 

Thank you for this feedback. We have added in additional information to the background. Authors have provided the relationship between components of the WHO-ICF and caregiver availability (page 5, and page 8). Given the gap in literature, our study will contribute to the body of knowledge related to WHO-ICF and caregiver availability. 

2. Please describe how the authors use the WHO-ICF model in caregiver availability well. 

We have provided how we applied WHO-ICF model in caregiver availability, drawing attention to how the caregiver availability among older adults is understood by us, within the framework of the WHO-ICF (highlighted in black in page 8).

3. Why did you use the WHO-ICF? Instead, you can use the social support theory.

Thank you for this suggestion. We employed WHO-ICF because we hypothesized that “perceived caregiver availability” for older adults may be related to the five components of WHO-ICF (personal factors, environmental factors, health condition, body function and structure, activity and participation). Social support theory would focus on one component of the ICF. Social support, including caregivers, would protect people from poor help and assist them in dealing with health events.

4. Please explore more about the relationship between physical disability and caregiver more in details

We have updated the information on relationship between physical disability and caregiver availability (bottom of page 5). During our literature review we identified that not much research has explored the relationship between physical disability and caregiver availability, which makes this study a novel one. 

5. What are your hypotheses?

We have orientated the reader to our hypothesis in text at the end of the introduction (bottom of page 8). Moreover, for each of the independent variables in the Method section, we have provided the expected relationship between each component and perceived caregiver availability (page 10- page 14). 

Methods.

1. What is the terminally ill? Please define it.

We have updated it as “seriously ill or unconscious “ and as determined by the health care staff. (page 9, under Study Sampling) 

2. Under study sample, Did you collect the data from the participants of the WHO on AGEing and adult health study? It is not clear.

The data was not collected from the participants of WHO SAGE study. I have updated the sentence as “This study sampled participants from a hospital setting, unlike the World Health Organisation study on AGEing and adult health (WHO-SAGE) that sampled older adults from their homes Kowal et al. (2012)” ( under Study Sampling on Page 9). 

3. Please include the references for the measurements. Some of them came from the articles. How about the rest?

I have provided reference of each of the independent variables (page 10-page 14). 

4. Data analysis. Please write more details about the software, code to obtain the odds ratios.

We have provided the detail as: “STATA version 15 was used to manage the analysis. And Because the analysis was logistic regression, we used logit with the dependent variable and the independent variables (i.e. logit dependent variable i.categorical independent variable continuous independent variable, or) (bolded black on page 14 under Data Analysis).”

5. The order of subsection in the method section is not correct.

We have reordered the subsections.

6. Please explain the relationships between the selected variables and subdomains in the WHOICF.

Thanks for the opportunity for the clarification. 

In the Method Section, we have specified the relationship of the selected variables and the subdomains of the WHO-ICF. We have also defined them to highlight how the variables selected are better for the subdomains. For instance, in the “activity limitation (AL)”, we have updated that “Activity limitation or disability refer to the difficulty older adults may experience in engaging activities necessary for life (World Health Organization, 2005). Based on previous finding that lower limb fracture have caregivers available to them (Li et al., 2017), we that older adults experiencing difficulty in engaging functional activities (disability), will have caregiver available to them.” (see page 10- page 14).

Discussion

1. The first paragraph is repetitive, like results. Please summarize it.

Thanks for your feedback. We have summarized the first paragraph of the discussion. In the first paragraph, we gave a summary of the applicability of the WHO-ICF framework to predicting perceived caregiver availability. This was important for us to establish the usefulness of the framework in the classification of factors affecting caregiver availability of older adults. 

2. Please exam your findings based on your hypotheses.

Since our hypothesis was in relationship to the WHO-ICF framework, we have discussed the finding in relations to the this framework. 

3. Since Ns of variables per domain is small. Instead of using the WHO-ICF, it would be better to examine the variable itself. Like marital status, chronic conditions. 

Thank you for your feedback. We appreciate there are a limited number of variables in each domain, and considered this in the discussion. We still feel that the WHO-ICF is a useful framework for considering how caregiver availability relates to needs.

4. Your findings are similar to previous studies. Please explain why and how it happened. 

Thanks for your feedback. We have discussed the findings in relation to the previous studies, with particular consideration on the application of the WHO-ICF framework. 

5. Please expand your conclusions to your unique area for the implication section.

We have expanded on the conclusion.(under conclusion on page 25).

Reviewer #2: This is an interesting manuscript that is worthy of publication.

The authors appreciate the acknowledgement. 

A few minor points that need to be addressed:

1. The first sentence of the second paragraph of the introduction does not make sense "...they need to do to live and those they most cherish." In addition, the reference that goes with this and the examples of self care is inappropriate (Shiel) and does not illustrate the point. 

Thanks for the feedback. Authors have corrected the sentence, and have cited a more appropriate reference most appropriate. The new reference cited shows examples of self-care activities including feeding, toileting etc. (bottom of page 3)

2. 3rd paragraph of the introduction - the Heaton reference is not listed.

The reference for Heaton has been provided (see page 4)

 3. Last paragraph of the introduction - the WHO 2002 references should read 2005.

We have provided the appropriate references. We used WHO, 2002 and WHO, 2005).(see page 7, last paragraph of the introduction) 

5. In the methods section, it is unclear as to whether the participants had to come back into the hospital to complete the survey, or if it was completed during their hospitalisation. If they had to come back, this is a bias because more participants with caregivers might be expected to have been able to arrange to come back than those without, i.e. the sample will be biased towards those with support.

Data was collected during participants’ hospitalisation. This clarification has been made in the work. (see page 10 under study sampling) 

6. In the results, in the third sentence of the paragraph 'Bivariate analysis' the sentence "Participants with no caregiver available had some no form of non-government support" does not make sense. 

We have revised the expression in the manuscript by deleting “some”. It is now read as “Participants with no caregiver available had no form of non-government support (100%)”.(under Bivariate analysis” on page 16)

7. In the paragraph commencing 'In Model 6...', the sentence "Participants living alone was not statistically significant" needs completing.

Revision to the sentence has been made in the manuscript. The sentence now read as “There was no statistically significant association between being single, separated or divorced and perceived caregiver availability (33% (p>0.05), compared to the married or cohabiting.”

8. Discussion last sentence of third paragraph repetition of the word 'modernisation'.

We have deleted one “Modernisation” in the last sentence of third paragraph. 

9. References: the Ghana statistical Service reference is not cited in the text.

Thanks for the feedback. However, Ghana Statistical Service reference is already cited in the text in the “Material and Method” under “Study area”. (page 9)

---

## [Editor Report · Decision Letter 1]

8 May 2020

Do factors across the World Health Organisation International Classification of Functioning, Disability and Health framework relate to perceived caregiver availability for community-dwelling older adults in Ghana?

PONE-D-19-33768R1

Dear Dr. Awuviry-Newton,

We are pleased to inform you that your manuscript has been judged scientifically suitable for publication and will be formally accepted for publication once it complies with all outstanding technical requirements.

With kind regards,

Frédéric Denis, Ph.D.

Academic Editor

PLOS ONE
---

## [Editor Report · Acceptance letter]

14 May 2020

PONE-D-19-33768R1 

Do factors across the World Health Organisation International Classification of Functioning, Disability and Health framework relate to perceived caregiver availability for community dwelling older adults in Ghana? 

Dear Dr. Awuviry-Newton:

I am pleased to inform you that your manuscript has been deemed suitable for publication in PLOS ONE. Congratulations! Your manuscript is now with our production department. 

With kind regards,

on behalf of

Dr. Frédéric Denis 

Academic Editor

PLOS ONE